# Seed Germination and Seedling Growth in *Suaeda salsa* (Linn.) Pall. (*Amaranthaceae*) Demonstrate Varying Salinity Tolerance among Different Provenances

**DOI:** 10.3390/biology12101343

**Published:** 2023-10-18

**Authors:** Wenwen Qi, Hongyuan Ma, Shaoyang Li, Haitao Wu, Dandan Zhao

**Affiliations:** 1Northeast Institute of Geography and Agroecology, Chinese Academy of Sciences, Changchun 130102, China; qiwenwen@iga.ac.cn (W.Q.); mahongyuan@iga.ac.cn (H.M.); lishaoyang@iga.ac.cn (S.L.); 2University of Chinese Academy of Sciences, Beijing 100049, China; 3Shandong Key Laboratory of Eco-Environmental Science for Yellow River Delta, Binzhou University, Binzhou 256603, China; zhaodandan@iga.ac.cn

**Keywords:** salinity stress, seed germination, seedling growth, interpopulation, ecological restoration, grassland

## Abstract

**Simple Summary:**

Soil salinization is a widespread problem, causing soil degradation and limiting agricultural productivity worldwide. However, the protection and restoration of halophyte species communities can effectively help improve and reconstruct saline–alkali land. We collected seeds of *Suaeda salsa* (Linn.) Pall., a highly salt-tolerant and widely distributed halophytic species, from seven different saline–alkali habitats, numbered S1–S7, of Songnen Plain, China, and tested their germination and seedling growth responses to saline–alkali stress. Results showed a strong saline–alkali tolerance in *S. salsa* seeds, but the germination characteristics and saline–alkali tolerance varied across provenances, highlighting the geographical variations in germination responses. The saline–alkali tolerance of S1 and S7 locations was significantly higher than that of others. *Suaeda salsa* has important applications in the restoration of saline–alkali soil. However, the seeds from appropriate provenances must be screened to ensure the best saline-alkaline tolerant variety. The germplasm materials from the selected varieties can also be used as high-quality sources to study the mechanism of saline–alkali tolerance in halophytes. These seeds can also serve as excellent germplasm resources for the restoration of saline–alkali soil.

**Abstract:**

Salinity is a pressing and widespread abiotic stress, adversely affecting agriculture productivity and plant growth worldwide. Seed germination is the most critical stage to seedling growth and establishing plant species in harsh environments, including saline stress. However, seed germination characteristics and stress tolerance may vary among geographical locations, such as various provenances. *Suaeda salsa* (Linn.) Pall. (*S. salsa*) is a halophytic plant that exhibits high salt tolerance and is often considered a pioneer species for the restoration of grasslands. Understanding the germination characteristics and stress tolerance of the species could be helpful in the vegetation restoration of saline–alkali land. In this study, we collected *S. salsa* seeds from seven different saline–alkali habitats (S1–S7) in the Songnen Plain region to assess the germination and seedling growth responses to NaCl, Na_2_CO_3,_ and NaHCO_3_, and to observe the recovery of seed germination after relieving the salt stress. We observed significant differences in germination and seedling growth under three salt stresses and among seven provenances. Resistance to Na_2_CO_3_ and NaHCO_3_ stress was considerably higher during seedling growth than seed germination, while the opposite responses were observed for NaCl resistance. Seeds from S1 and S7 showed the highest tolerance to all three salt stress treatments, while S6 exhibited the lowest tolerance. Seeds from S2 exhibited low germination under control conditions, while low NaCl concentration and pretreatment improved germination. Ungerminated seeds under high salt concentrations germinated after relieving the salt stress. Germination of ungerminated seeds after the abatement of salt stress is an important adaptation strategy for black *S. salsa* seeds. While seeds from most provenances regerminated under NaCl, under Na_2_CO_3_ and NaHCO_3_, only seeds from S4 and S7 regerminated. These findings highlight the importance of soil salinity in the maternal environment for successful seed germination and seedling growth under various salinity-alkali stresses. Therefore, seed sources and provenance should be considered for vegetation restoration.

## 1. Introduction

Soil salinization is a widespread environmental issue limiting agricultural production worldwide [1]. According to FAO/UNESCO, globally, saline soil occupies approximately 1 billion hm^2^ of land, accounting for 7.2% of the world’s total land area [2,3]. Recently, the interactive effects of climate change and anthropogenic activities have led to a decline in plant biodiversity and productivity, posing significant challenges to the management and restoration of saline–alkali land [4,5]. However, many methods have been implemented to improve this land, including irrigation and drainage [6], application of soil amendment [7], and planting salt-tolerant plants [5]. Particularly, planting saline–alkaline tolerant plants effectively reduces soil salt content and improves soil structure. These plants save water, require low input, and provide environmental sustainability and economic benefits, making them a popular research object for studying saline–alkali land [8]. Therefore, halophytic plants have received considerable attention in vegetation restoration on saline–alkaline lands, as most other plants struggle to grow in these soil types. Indeed, several studies have examined the saline–alkali tolerance of halophytes [9,10,11]. Therefore, protecting and rebuilding halophyte plant communities is an effective method to improve and restore saline–alkali land [12,13].

Saline–alkali stress can have a negative impact on plant growth, development, and reproduction through osmotic stress and ion toxicity. However, halophytes are characterized by their ability to reject, accumulate, and absorb salt, allowing them to thrive and grow in saline–alkali environments [14]. In addition, some saline–alkali tolerant plants can complete their life cycle in soil with salinity levels above 200 mmol/L [15,16]. Previous studies have shown that seed germination and seedling growth are critical stages and particularly susceptible to salt stress, affecting plants’ ability to establish themselves in saline–alkali environments [17]. Hence, understanding the stress resistance of halophytes in the early stages of their life cycle and under saline–alkali stress will improve the application of halophytes for vegetation restoration in saline–alkali lands [18,19]. However, germination characteristics and stress tolerance of halophytic species vary among provenances, likely due to local adaptation or transgenerational induction through epigenetic effects of maternal environments [20,21,22]. For instance, *Plantago lanceolata* L. (*Plantaginaceae*) [23], *Prosopis Juliflora* (Sw.) DC. *(Fabaceae)* [9], *Arthrocnemum macrostachyum* (Moric.) C. Koch (*Amaranthaceae*), and *Suaeda vermiculat* Forssk. ex J.F. Gmel. [10] are common halophytes that differ significantly in their seed germination and ability to recover after saline–alkali stress amelioration across provenances. These differences are likely related to environmental factors within the particular provenance, including the heterogeneity of saline–alkali soils [24,25]. Therefore, understanding the germination behaviors of seeds from different populations can inform the selection of optimal best seed sources to improve and restore saline–alkali lands.

As a euhalophyte, *Suaeda salsa* (Linn.) Pall. (*S. salsa*) is highly salt-tolerant and widely distributed in coastal salt marshes and inland salt-bearing locations. This species is also used in animal feed, gourmet vegetables, and edible oil, with high ecological and economic values [26,27]. There are two main *S. salsa* ecotypes: the green ecotype, which typically grows at high elevations and on land far from the seashore, and the red-violet ecotype, which grows in the intertidal zone [28]. *Suaeda salsa* produces dimorphic (black and brown) seeds [29]. This species can accumulate a high amount of salt in its aboveground portions [30] and demonstrate salt tolerance characteristics, such as ion compartmentalization and succulent leaves [31,32]. Hence, *S. salsa* is considered a pioneer species for restoring saline–alkali lands [33]. Understanding the seed germination characteristics and stress tolerance of *S. salsa* will be particularly helpful for vegetation restoration in saline–alkali lands. However, most studies have focused on coastal saline–alkali land areas [34,35,36], where species salt tolerance capacity has been tested primarily using neutral salts, such as sodium chloride (NaCl) and sodium sulfate (Na_2_SO_4_). Meanwhile, few studies have considered alkaline salts [37,38,39], with minimal data regarding the stress tolerance capacity of *S. salsa* growing in inland soda saline–alkali soil.

The Songnen Plain, located in the northeast plain of China, is an ecotone with a high concentration of saline–alkaline soil [40]. The plain occupies approximately 5 × 10^6^ hm^2^ of land, with 64% of the grassland area in the western part of the Songnen Plain degraded, resulting in a saline–alkali wasteland [41]. Particularly, the saline–-alkali soil with sodium carbonate (Na_2_CO_3_) and sodium bicarbonate (NaHCO_3_) as the main salts [42] has high pH, salt, and exchangeable sodium content. However, the soil has low permeability and nutrient availability, which differs from the coastal saline soil that contains primarily NaCl or Na_2_SO_4_ [43]. Indeed, alkaline stress can be more damaging to plants than neutral salt stress [44,45,46,47].

Halophyte salinity tolerance is usually species-specific and varies with ecotype and the environmental factors of the provenances [10,48]. Therefore, to explore the potential differences in the salinity and alkali tolerance in *S. salsa* seeds across the provenances, we collected *S. salsa* seeds from seven different saline–alkali habitats of the Songnen Plain. Specifically, we (1) investigated variations among the different seed sources regarding seed germination and seedling growth and (2) tested the seed germination and seedling growth responses to NaCl, Na_2_CO_3_, and NaHCO_3_. Results from this study may help identify salt-tolerant germplasm materials, which can be utilized for vegetation restoration in the saline–alkali region of the Songnen Plain.

## 2. Materials and Methods

### 2.1. Seed Collection

Seeds from multiple *S. salsa* plants were collected from 7 different provenances of the Songnen Plain in northeastern China (Figure 1) in September 2018. In each provenance, seeds were collected from 30 randomly chosen green ecotype parent plants at least 5 m apart to reduce potential pseudoreplication. The populations were labeled as ‘S’ followed by a number identifying the specific site (Figure 1). Seeds were immediately stored in paper bags at 4 °C until the experiment began in November 2019. General features and environmental variables from each provenance are listed in Table 1.

### 2.2. Seed Germination, Seedling Growth, and Germination Recovery

Healthy and plumb black *S. salsa* seeds were selected and surface sterilized in an aqueous solution of 0.1% HgCl_2_ for 10 min, then rinsed with distilled water three times before running the germination experiments [49]. We constructed six NaCl (50, 100, 200, 300, 400, 500 mM), five Na_2_CO_3_ (5, 12.5, 25, 50, 100 mM), and four NaHCO_3_ (25, 50, 100, 150 mM) solutions. In each experiment, 25 seeds were placed in each 10-cm diameter Petri dish filled with 25 mL of germination medium containing 7% agar and different salt concentrations. A non-saline germination medium was used as the control. Each treatment was replicated four times. A total of 100 seeds (25 seeds × 4 replicates = 100) were used per treatment.

Petri dishes were incubated (Harbin, China) at 16/28 °C under a 12/12 h dark/light diurnal cycle [50,51]. Seed germination was considered when the radicle emerged. The experiment was stopped when no new germination was observed for three consecutive days. The germinated seeds were counted once per day, and the germination rate was calculated using the following formula [52]:Germination=n/N×100%,
where n is the number of final germinated seeds under each treatment, and N is the total number of seeds.

Once the germinated seeds formed seedlings, 4 seedlings were collected from each Petri dish to measure the roots and shoot lengths and determine the root and shoot stress tolerance index using the following formulas [52]:Root tolerance index (%)=rl/crl×100%
Shoot tolerance index (%)=sl/csl×100%,
where rl and sl are the root and shoot lengths in different salt treatments, respectively, and crl and csl are the root and shoot lengths under 0 mM salt stress treatment, respectively.

After the initial germination experiment, we established the germination recovery experiment. Ungerminated seeds from each treatment were rinsed three times with distilled water and incubated in new Petri dishes with the same germination medium and thermo period regimes. The germinated seeds were counted after 7 days to determine the percentage of germination recovery. We also calculated total and relative germination percentages using the following formulas [52]:$$Recovery\;percentage\;\left(\%\right)=\left(Rn/Un\right)\times 100\%$$
Total germination percentage (%)=Tn/N×100%
Relative germination percentage (%)=(Tn/Cn)×100%,
where Rn represents the number of germinated seeds in the recovery experiment, Un is the number of ungerminated seeds, Tn is the sum of n and Rn, N is the total number of seeds, and Cn is the number of germinated seeds under 0 mM salt stress.

### 2.3. Statistical Analysis

Generalized linear models (GLMs) with binomial error and the log link function were used to evaluate the effects of the different treatments on percent germination, germination recovery, and total germination [52]. Linear regression models were used to analyze the effects of different treatments on root and shoot length and relative percent germination. Post-hoc multiple mean comparisons were performed with Tukey’s tests. A cluster analysis was performed to visualize the patterns of salt tolerance in *S. salsa* seeds across 7 provenances and salt levels [50]. Results were considered statistically significant at *p* = 0.05. All statistical analyses were performed in R version 4.0.3, and the graphs were created using ggplot.

## 3. Results

### 3.1. Effect of Provenance on Seed Germination under NaCl Stress

There was no significant difference in seed germination when the salt concentration was below 300 mM (Figure 2). In contrast, low salt concentration promoted the elongation of roots and shoots (Figure 3 and Figure 4). When the salt concentration was ≥300 mM, the germination rates and seedling growth were significantly inhibited (Figure 2, Figure 3 and Figure 4); compared to shoots, the salt tolerance index of the roots was lower, suggesting roots were more sensitive to high salt stress (Table 2 and Table 3). However, after the high salt stress was relieved, most ungerminated seeds germinated (Figure 5, Table 4).

A significant relationship was observed between the provenances and NaCl concentrations in *S. salsa* seed germination (Table 5). Under high NaCl stress, the salt tolerance of seeds from S5 and S7 was significantly higher than that of other provenances, with 27% and 15% germination rates under 500 mM NaCl, respectively. However, the seeds from S1, S2, and S6 did not germinate, while only one seed germinated from S3 and S4 (Figure 2G). When the salt concentration reached ≥ 300 mM, germination in S1, S3, and S4 decreased significantly, with rates that were > 50% lower than the control (Figure 2E–G). Hence, 300 mM may represent the salt-tolerance threshold for *S. salsa* during germination in these three provenances.

For each recovery treatment, germination, total germination, and the relative germination rates of S2 and S6 were lower than those of other provenances (Figure 2 and Figure 5, Table 6). These results suggest that the seeds from S2 and S6 may be in a dormant period. In 50–200 mM NaCl, total germination and the relative germination rates in S2 were higher than in the control (Figure 5, Table 6). Meanwhile, under 200–500 mM NaCl, total S2 germination was significantly higher than S6 (Figure 5), indicating that seeds from S6 were most sensitive to high salt stress.

Moreover, seedling growth was significantly different across provenances. In each treatment, seedling growth in S4 and S5 was markedly faster than in other provenances; the slowest growth was observed in S6. Under 50–300 mM NaCl in S7, the shoot length was significantly higher than that of the control. In addition, the shoot tolerance index of S7 was significantly higher than in other provenances (Table 4).

### 3.2. Effect of Provenances on Seed Germination under Na_2_CO_3_ Stress

*S. salsa* seeds showed weak germination response to Na_2_CO_3_ treatments in all seven provenances (Figure 6). However, seedling growth was not hindered by Na_2_CO_3_ stress (Figure 7 and Figure 8). At 5 mM Na_2_CO_3_, the *S. salsa* germination rate decreased by more than 50% compared to the control (Figure 6), and seed germination progressively decreased with increasing Na_2_CO_3_ concentration. In addition, the total germination percentage in the recovery experiment was significantly lower than in the control (Figure 9 and Table 7). However, the root length under 5–25 mM Na_2_CO_3_ and the shoot length across all Na_2_CO_3_ treatments was significantly higher than the control (Figure 7 and Figure 8). The shoot stress tolerance index under 5–25 mM exceeded the root stress tolerance index. In contrast, the shoot stress tolerance index under 50–100 mM was higher than the root stress tolerance index, suggesting that roots are more sensitive to Na_2_CO_3_ stress (Table 8 and Table 9).

Significant relationships were observed between the provenances and Na_2_CO_3_ concentrations in *S. salsa* germination and seedling growth (Table 10). The germination rate of S1 was higher than other provenances across all Na_2_CO_3_ concentrations, with S2 showing the lowest germination (Figure 6). However, the root length of S2 and shoot length of S2, S4, and S5 were significantly higher than other provenances, and the root and shoot lengths of S6 were the lowest (Figure 7 and Figure 8). Germination recovery showed no significant difference across the provenances or Na_2_CO_3_ concentrations, except S7, where percent germination was higher than the control across all Na_2_CO_3_ treatments (Figure 9 and Table 11).

### 3.3. Effect of Provenance on Seed Germination under NaHCO_3_ Stress

Seed germination was significantly inhibited cross all NaHCO_3_ treatments, excluding S1 (Figure 10). We found significant relationships between provenance and NaHCO_3_ concentration regarding *S. salsa* germination and seedling growth (Table 12). However, seedling growth was less inhibited under NaHCO_3_ than under Na_2_CO_3_ treatments (Figure 7, Figure 8, Figure 11 and Figure 12, Table 10, Table 11, Table 13 and Table 14). Root length was not inhibited under any NaHCO_3_ treatment, except for S6; root length was longest in S2, while those in S2 and S5 were significantly longer than other provenances under each NaHCO_3_ treatment. The shoot stress tolerance indices under all NaHCO_3_ treatments were significantly higher than under the Na_2_CO_3_ treatments, except for S4.

Regarding germination recovery, all provenances, excluding S2 and S6, exhibited a relapse phenomenon. The germination recovery for S7 was the highest (Table 15), although total germination was significantly lower than that of control (Figure 13 and Table 16). No significant difference was observed in seed germination between S7 and S1 (Figure 13). Thus, NaHCO_3_ also promoted seedling growth. Across the provenances, S1 showed the highest stress resistance, followed by S7; meanwhile, NaHCO_3_ significantly inhibited germination in all other provenances.

### 3.4. Cluster Analysis of Seed Germination Data

Cluster analysis was performed using germination data to comprehensively understand differences in saline–alkali tolerance among seeds from different provenances. The seven provenances were classified into three groups (Figure 14): Group I included S3, S4, S5, and S7; Group II included S1; Group III included S2 and S6. Cluster analysis and regression analyses suggested that seeds from S3, S4, S5, and S7 showed the highest NaCl tolerance. Moreover, seeds from S1 had moderate NaCl tolerance and the highest NaHCO_3_ tolerance. In contrast, seeds from S2 and S6 had the lowest NaCl, Na_2_CO_3_, and NaHCO_3_ tolerance.

Based on the sampling sites’ geographical distribution and environmental conditions, S3, S6, and S7 were close to each other, with similar climatic conditions. Meanwhile, S1, S2, S4, and S5 were distant. However, S4 and S5 had similar climatic conditions despite being far apart. In contrast, the differences in soil conditions based on geographical distance and climatic conditions were inconsistent. For instance, the soil pH at the S1, S2, and S4 sampling sites was higher, while the soil ion content was lower in S2. In contrast, soil pH, EC, and ion content at sampling site S6 were the lowest. Additionally, soil from S3, S4, S5, and S7 had the highest EC and Cl^−^ content, while that from S4 and S7 had the highest HCO_3_^−^ content. Overall, our results suggest that the differences in seed germination among different provenances may be attributed to local edaphic factors rather than geographical locations and climatic conditions.

## 4. Discussion

Previous studies indicate alkali stress is more destructive to plants than neutral salt stress. Under alkaline stress, seed germination is affected primarily by osmotic imbalance with high pH [53,54]. Consistent with this, in the current study, Na_2_CO_3_ and NaHCO_3_ had stronger inhibitory effects on *S. salsa* germination than NaCl, regardless of provenance, excluding S1. After the saline–alkali stress was relieved, ungerminated seeds from all provenances except S6 and under 200–500 mM NaCl showed comparable responses to other halophytes [55,56,57]. However, seeds from S7 and under 25–150 mM NaHCO_3_ regerminated. Regermination is an important strategy for *S. salsa* black seeds to adapt to salinity stress, in which seeds remain viable at high salinity and germinate when environmental conditions become favorable [29,58]. Previous studies involving regermination have shown that germination inhibition is likely due to osmotic restriction rather than ionic toxicity [11,29]. In this study, the stronger inhibition of seed germination by Na_2_CO_3_ and NaHCO_3_ than NaCl is likely due to *S*. *salsa* seeds being inactive or dormant under the combined stress of high pH, osmotic pressure, and ionic toxicity. 

Contrary to the seed germination responses, 5–50 mM Na_2_CO_3_ and 25–150 mM NaHCO_3_ promoted root and shoot growth in *S. salsa*. Indeed, plants have different levels of abiotic stress resistance at different growth stages [19,59]. *Suaeda salsa*, as a halophyte, can tolerate salt stress at the early stages of growth, and low salinity levels can promote seedling growth [27,34,60]. We found greater Na_2_CO_3_ and NaHCO_3_ tolerance in *S. salsa* during the seedling growth stage than in the seed germination stage. In contrast to our findings, Li et al. [19] found enhanced seedling growth under 200 mM NaCl, while Na_2_CO_3_ and NaHCO_3_ significantly inhibited germination and seedling growth. 

In our study, root growth in S6 was significantly inhibited under NaCl, Na_2_CO_3,_ and NaHCO_3_ stress, which differed from what was observed for other provenances. In addition, germination percentages in S2 and S6 were significantly lower than those of other provenances under control treatment, while seedling growth in S2 was significantly higher than in S6. These results show that besides halophyte characteristics, other factors impact their tolerance to saline–alkali conditions. Germination, seedling growth, and abiotic stress tolerance in many plant species vary spatially (i.e., provenances), likely due to local adaptations or transgenerational induction through epigenetic effects of maternal environments, such as precipitation, temperature, and soil nutrients [20,61,62]. However, our cluster analysis results showed no apparent changes in seed germination or seedling growth along temperature and precipitation gradients; rather, they exhibited a strong correlation with the soil properties of the provenances. Therefore, different germination and seedling growth strategies in *S. salsa* from different provenances and under different saline–alkali stresses may be explained by a long-term adaptation of the parent plant (i.e., seed source) to the local soil environment. Since the maternal environment of seed sources plays an important role in determining the germination characteristics of seeds, it is vital to consider provenances to collect seeds for vegetation restoration in Songnen Plain.

Under high salt stress, the germination rates were high, while seedling growth was significantly inhibited, increasing the risk of plant establishment in saline–alkali land [30,63]. In our study, S5 and S6 were not suitable for heavy saline–alkali soil dominated by NaCl, S7 was suitable for Na_2_CO_3_, and S1 and S7 were suitable for NaHCO_3_. Previous studies suggest significant differences in seed germination in two ecotypes and dimorphic seeds of *S. salsa* when exposed to saline–alkali stress. Black seeds have a more extended dormancy period and lower germination rate than brown seeds. However, the germination rate of the black seeds from *S. salsa* plants growing at high elevations and farther inland is higher than those growing in the intertidal zone [28,29]. Meanwhile, in this study, we only selected black seeds collected from the same ecotype, with very few brown seeds. Therefore, the intraspecific variation in different ecotypes and dimorphic seeds of *S. salsa* growing in soda saline–alkali land in Songnen Plain needs further study.

## 5. Conclusions

Our study revealed significant differences in germination and seedling growth in *S. salsa* under various salt stresses and seven provenances. NaCl, Na_2_CO_3,_ and NaHCO_3_ demonstrated varying degrees of inhibition on seed germination. However, salt resistance in *S. salsa* was significantly higher under Na_2_CO_3_ and NaHCO_3_ during seedling growth than during germination. The roots appeared to be more sensitive to salt stress than the shoots, suggesting roots can be used as an effective indicator to evaluate salt tolerance in *S. salsa.* Our study also provides evidence that the soil salinity of the maternal environment (i.e., provenances in this study) is an important environmental factor affecting salt tolerance in seeds. Under various concentrations of three salts stress, seeds from S1 and S7 were the most salt and alkali tolerant among all provenances, while those from S6 were the least stress tolerant. Seeds from S2 showed low germination under the control condition, while low NaCl concentrations and pretreatments enhanced their germination. This study provides basic information for cultivating and protecting high-quality saline–alkali resistant species, which can be utilized for saline–alkali land reconstruction and vegetation restoration in the Songnen Plain.

## Figures and Tables

**Figure 1 biology-12-01343-f001:**
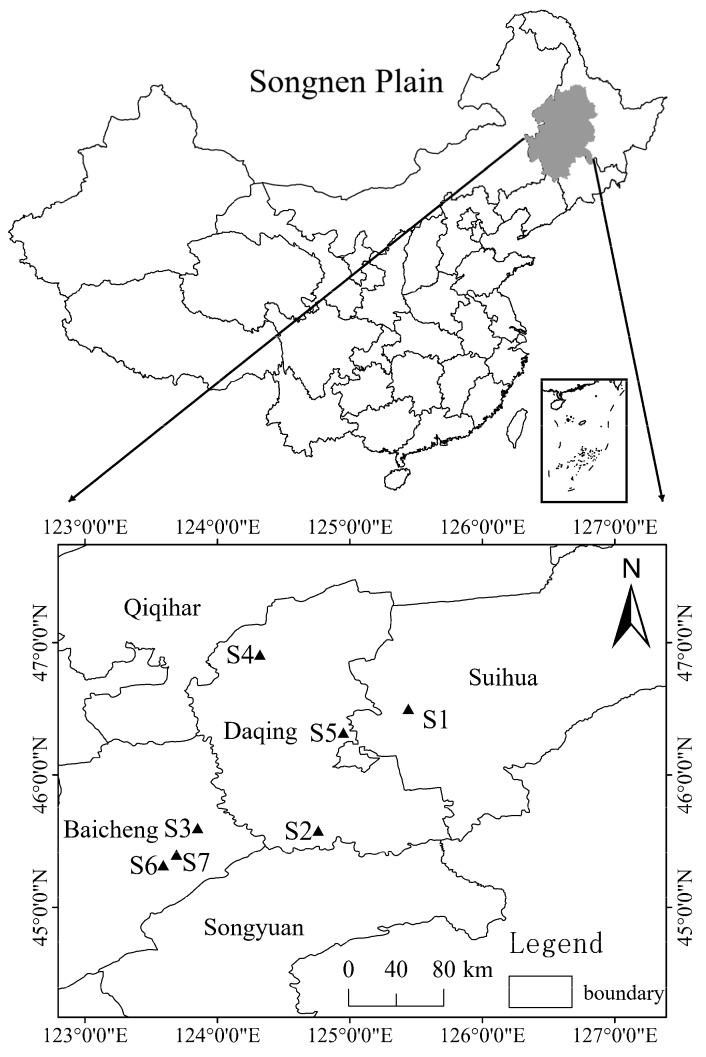
A location map of the study area showing S1–S7 provenances in the Songnen Plain. Note: Baicheng, Qiqihar, Daqing, Songyuan, and Suihua are the names of the cities where the sampling points were located.

**Figure 2 biology-12-01343-f002:**
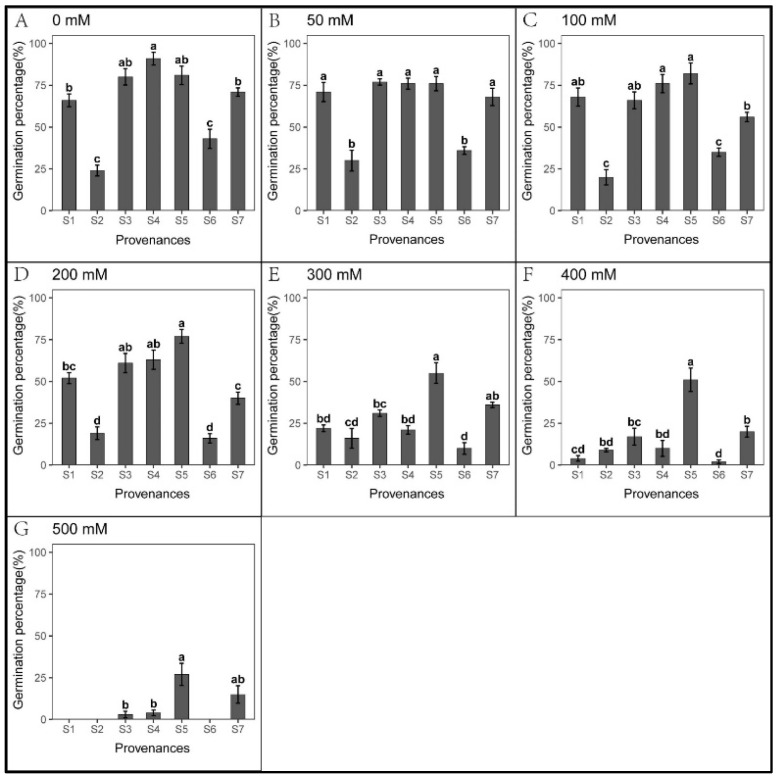
Mean (±SE; n = 4) percent seed germination of seven provenances in response to different NaCl concentrations. S1–S7 represent the seven provenances; (**A**–**G**) NaCl concentrations (0–500 mM). Bars with different letters represent significant differences in seed germination among the treatments and provenances (*p* < 0.05, Tukey’s test).

**Figure 3 biology-12-01343-f003:**
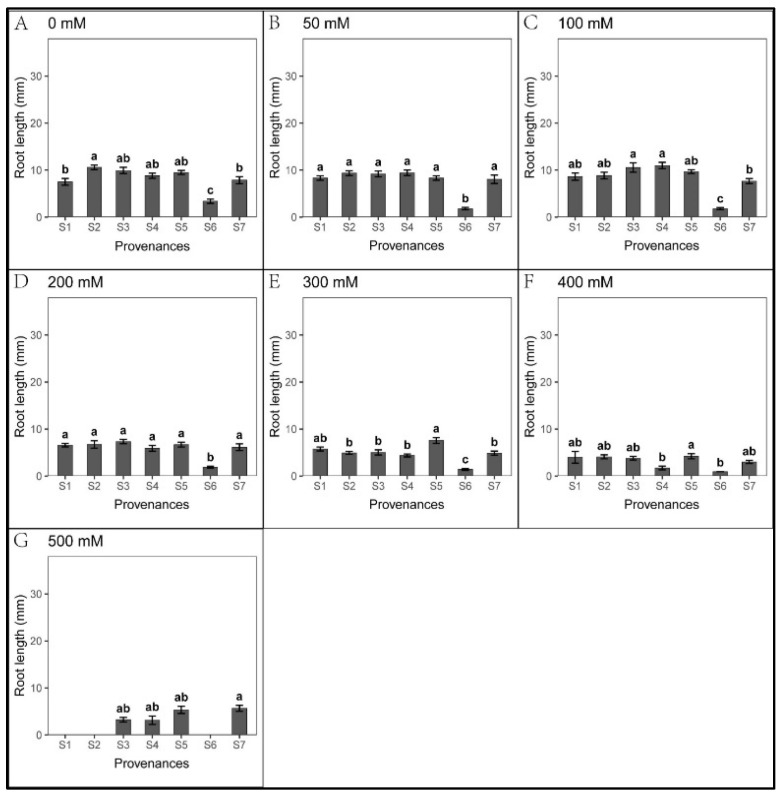
Mean (±SE; n = 16) root length of seven provenances in response to different NaCl concentrations. S1–S7 represent the seven provenances; (**A**–**G**) NaCl concentrations (0–500 mM). Bars with different letters represent significant differences in root length among the provenances (*p* < 0.05, Tukey’s test).

**Figure 4 biology-12-01343-f004:**
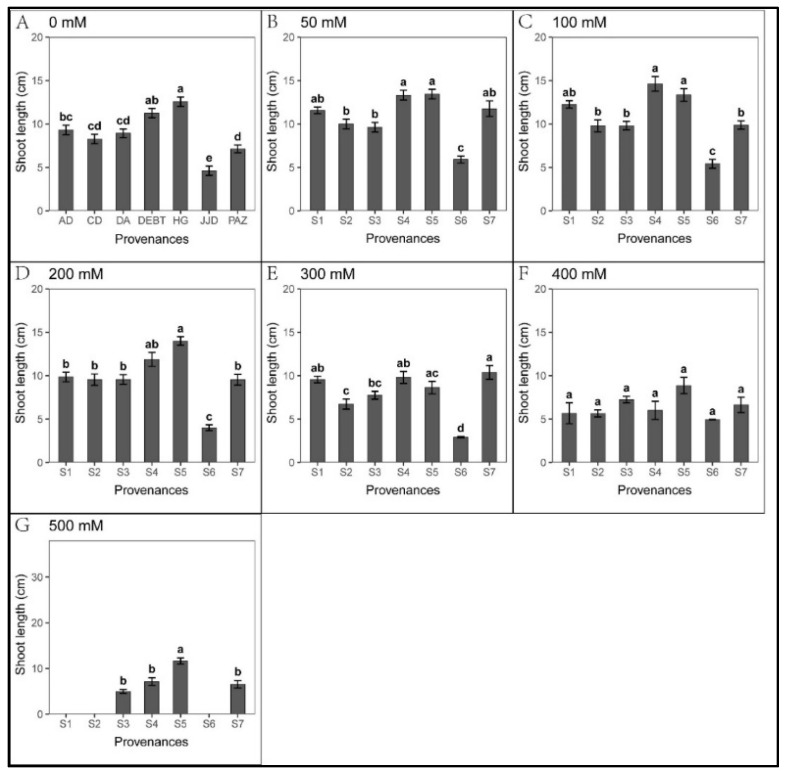
Mean (±SE; n = 16) shoot length of seven provenances in response to different NaCl concentrations. S1–S7 represent the seven provenances; (**A**–**G**) NaCl concentrations (0–500 mM). Bars with different letters represent significant differences in shoot length among the provenances (*p* < 0.05, Tukey’s test).

**Figure 5 biology-12-01343-f005:**
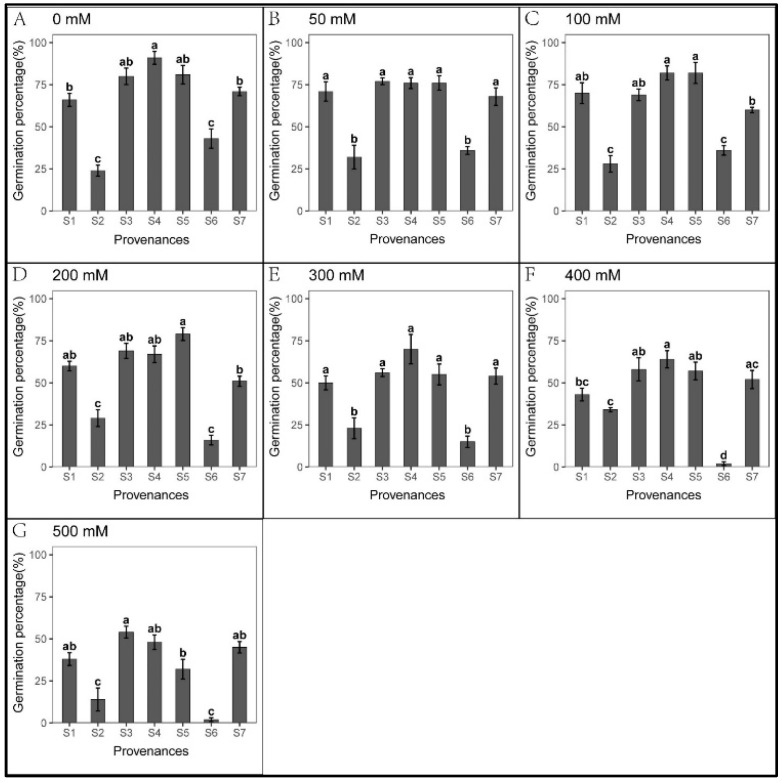
Mean (±SE; n = 4) percent total germination of seven provenances in response to different NaCl concentrations. S1–S7 represent the seven provenances; (**A**–**G**) NaCl concentrations (0–500 mM). Bars with different letters represent significant differences in total germination percentage among the provenances (*p* < 0.05, Tukey’s test).

**Figure 6 biology-12-01343-f006:**
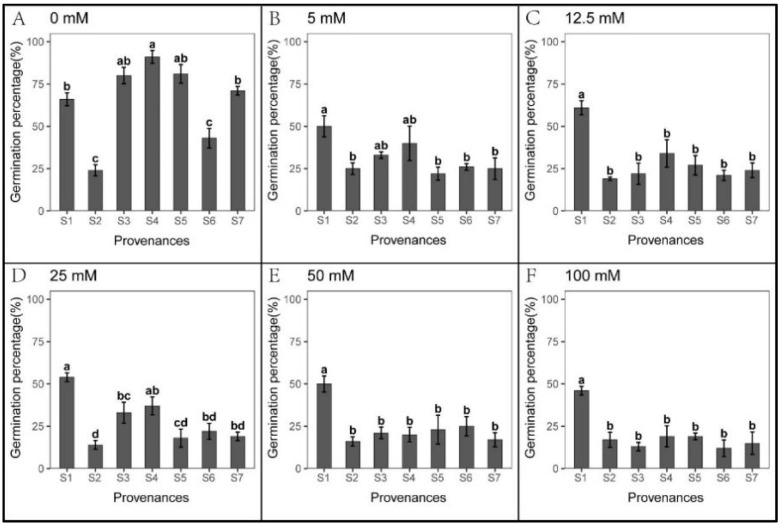
Mean percent (±SE; n = 4) seed germination across seven provenances in response to different Na_2_CO_3_ concentrations. (**A**–**F**) Na_2_CO_3_ concentrations (0–100 mM). Bars with different letters represent significant differences among the provenances (*p* < 0.05, Tukey’s test).

**Figure 7 biology-12-01343-f007:**
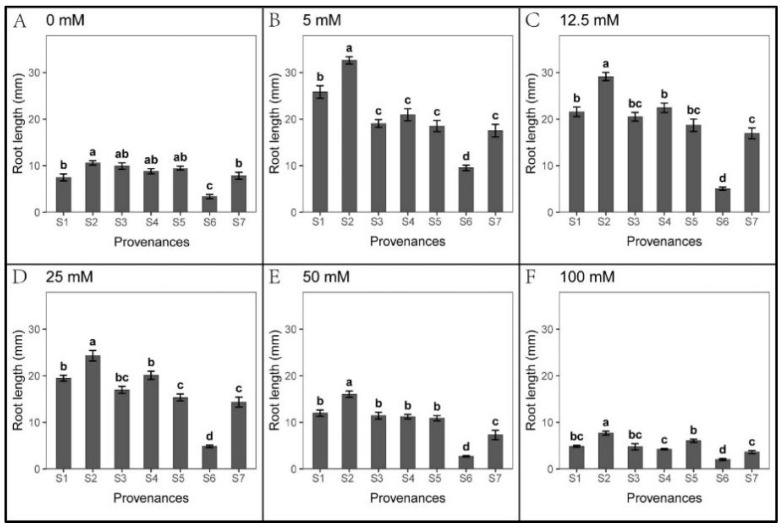
Mean root length (±SE; n = 16) of seven provenances in response to different Na_2_CO_3_ concentrations. S1–S7 represent the seven provenances; (**A**–**F**) Na_2_CO_3_ concentrations (0–100 mM). Bars with different letters represent significant differences among the provenances (*p* < 0.05, Tukey’s test).

**Figure 8 biology-12-01343-f008:**
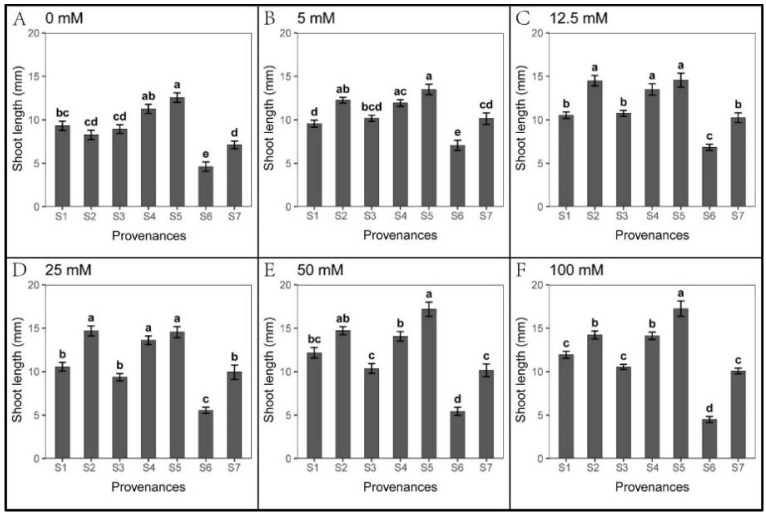
Mean shoot length (±SE; n = 16) of seven provenances in response to different Na_2_CO_3_ concentrations. S1–S7 represent the seven provenances; (**A**–**F**) Na_2_CO_3_ concentration (0–100 mM). Bars with different letters represent significant differences among the provenances (*p* < 0.05, Tukey’s test).

**Figure 9 biology-12-01343-f009:**
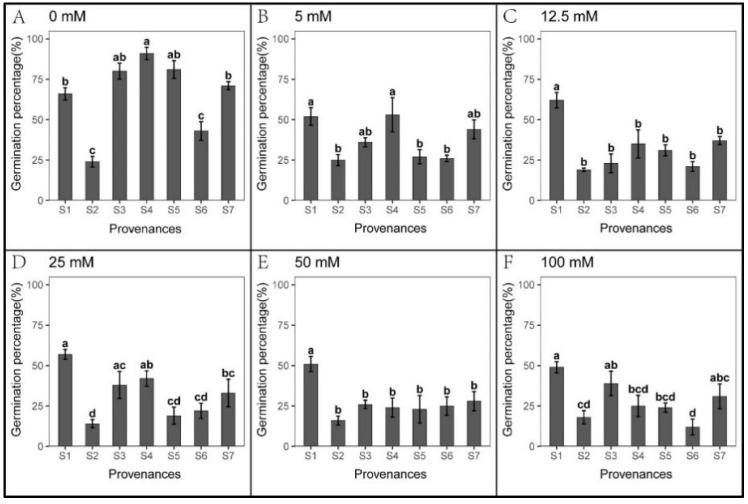
Mean (±SE; n = 4) total germination percentage of seven provenances in response to different Na_2_CO_3_ concentrations. S1–S7 represent the seven provenances; (**A**–**F**) Na_2_CO_3_ concentrations (0–100 mM). Bars with different letters represent significant differences among the provenances (*p* < 0.05, Tukey’s test).

**Figure 10 biology-12-01343-f010:**
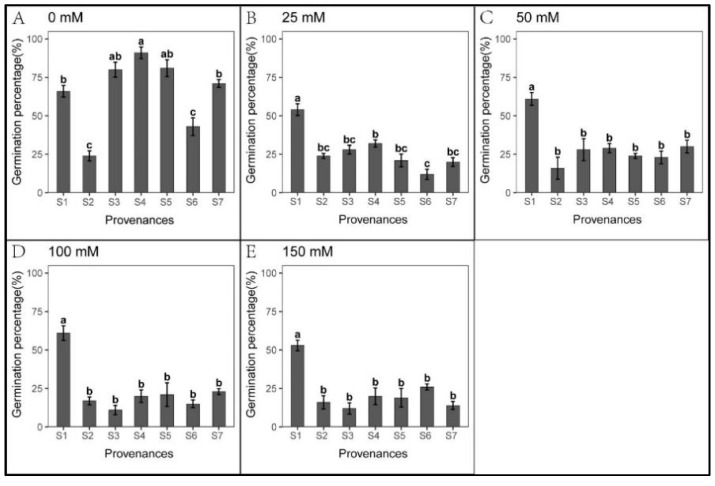
Mean (±SE; n = 4) percent seed germination across seven provenances in response to different NaHCO_3_ concentrations. S1–S7 represent the seven provenances; (**A**–**E**) NaHCO_3_ concentrations (0–150 mM). Bars with different letters represent significant differences in percent germination among the provenances (*p* < 0.05, Tukey’s test).

**Figure 11 biology-12-01343-f011:**
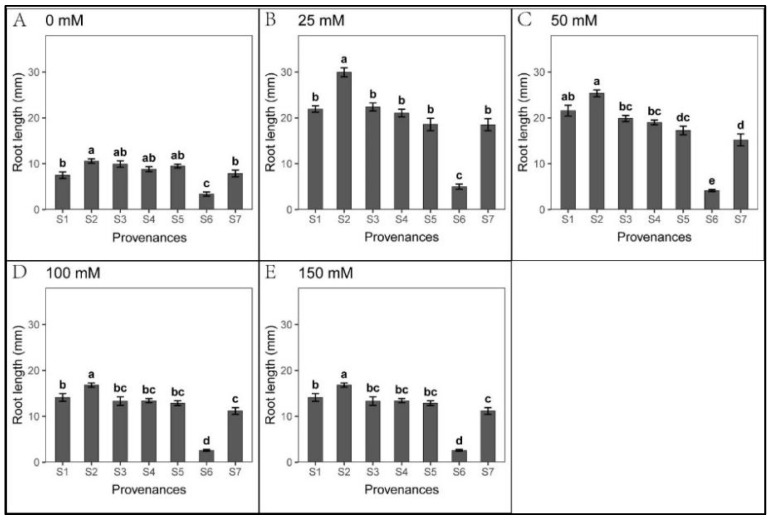
Mean root length (±SE; n = 16) of seven provenances in response to different NaHCO_3_ concentrations. S1–S7 represent the seven provenances; (**A**–**E**) NaHCO_3_ concentrations (0–150 mM). Bars with different letters represent significant differences in root length among the provenances (*p* < 0.05, Tukey’s test).

**Figure 12 biology-12-01343-f012:**
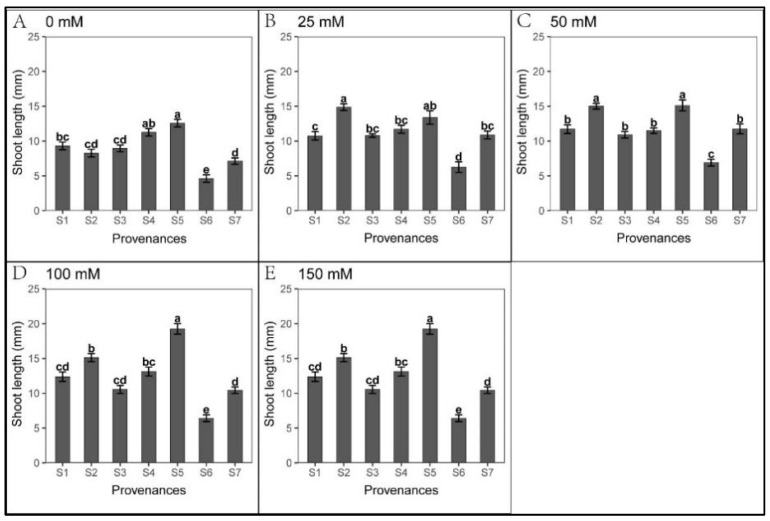
Mean shoot length (SE; n = 16) of seven provenances in response to different NaHCO_3_ concentrations. S1–S7 represent the seven provenances, (**A**–**E**) NaHCO_3_ concentrations (0–150 mM). Bars with different letters indicate significant differences in shoot length among the provenances (*p* < 0.05, Tukey’s test).

**Figure 13 biology-12-01343-f013:**
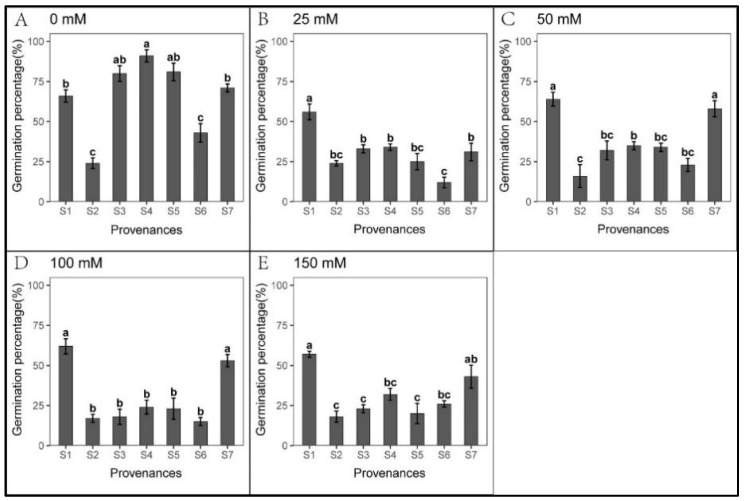
Mean (SE; n = 4) total percent seed germination of seven provenances in response to different NaHCO_3_ concentrations. S1–S7 represent the seven provenances; (**A**–**E**) NaHCO_3_ concentrations (0–150 mM). Bars with different letters indicate significant differences in total seed germination among the provenances (*p* < 0.05, Tukey’s test).

**Figure 14 biology-12-01343-f014:**
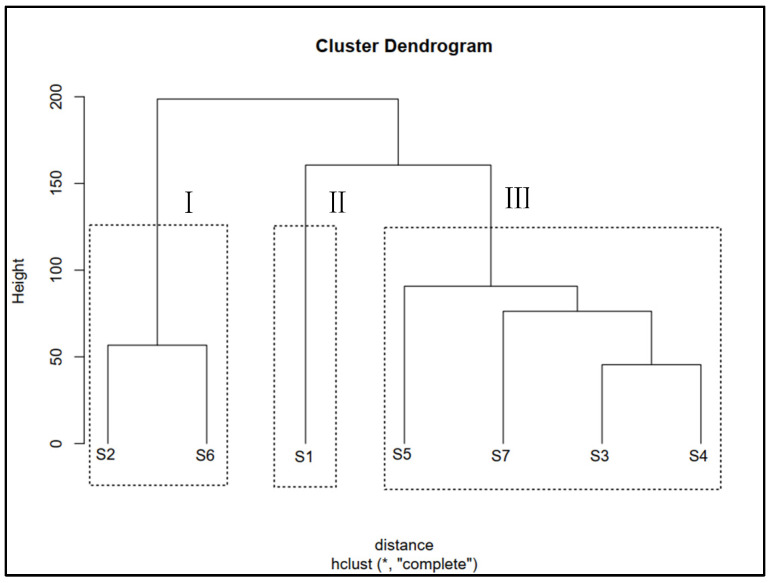
Cluster analysis of the S1–S7 sites based on seed germination rates at different salinity levels. I, II, and III represent the three salt tolerance classifications.

**Table 1 biology-12-01343-t001:** General features and environmental variables for S1–S7 provenances. Climate data were acquired from https://www.tianqi24.com/historycity/ (accessed on 8 August 2022).

Provenance	Latitude	Longitude	MAT (°C/year)	MAP (mm/year)	pH	EC (mS/cm)	Na^+^ mmolc/L	Cl^−^ mmolc/L	CO_3_^2−^ mmolc/L	HCO_3_^−^ mmolc/L
S1	46.50	125.44	5	535.32	10.39	0.830	16.63	1.05	0.60	13.20
S2	45.58	124.76	4	924.09	10.16	1.065	17.18	4.50	4.00	4.60
S3	45.60	123.85	6	559.7	9.50	2.320	21.30	23.10	3.80	15.20
S4	46.91	124.32	10	338.06	10.45	2.800	25.31	17.25	0.20	23.20
S5	46.32	124.95	9	354.33	9.43	2.500	23.25	24.56	2.30	10.23
S6	45.32	123.59	6	559.7	8.74	0.087	2.23	0.90	0.10	4.80
S7	45.40	123.69	6	559.7	9.52	2.260	19.30	13.22	2.50	23.60

Note: MAT = mean annual temperature; MAP = mean annual precipitation; EC = soil electric conductivity. pH, EC, and soil ion content data were obtained by measuring and analyzing 0–20 cm topsoil.

**Table 2 biology-12-01343-t002:** Root salt tolerance index of *S. salsa* seedlings grown from seeds treated with different NaCl concentrations across the provenances. Each treatment had 16 biological replicates. Different lower-case letters indicate significant differences (*p* < 0.05, Tukey’s test) between treatments within the same provenance, while different upper-case letters indicate significant differences (*p* < 0.05, Tukey’s test) between provenances under the same NaCl concentration.

Concentration (mM)	S1	S2	S3	S4	S5	S6	S7
0	100 ± 0 Aab	100 ± 0 Aa	100 ± 0 Aab	100 ± 0 Ab	100 ± 0 Aa	100 ± 0 Aa	100 ± 0 Aac
50	112.3 ± 7.1 Aa	89.3 ± 1.9 Aa	93.3 ± 3.6 Ab	107.8 ± 3.2 Aab	87.3 ± 1.7 Aab	42.2 ± 7.4 Bb	103.5 ± 9.8 Aa
100	113.7 ± 8.9 Aa	83.3 ± 2.5 BCa	106.7 ± 2.7 ABa	126.5 ± 7.2 Aa	102 ± 2.6 ABa	58.9 ± 11 Cb	100.7 ± 6.9 ABab
200	87.6 ± 1.4 Aab	58.2 ± 8.6 Bb	74.7 ± 3.6 ABc	65.9 ± 4.3 ABc	70.1 ± 2.8 ABbc	62.1 ± 4.2 Bb	76.4 ± 4.8 ABbd
300	76.3 ± 4.9 Abc	43.6 ± 2.6 Cb	51.1 ± 2.1 BCd	49.7 ± 2.4 BCc	79.5 ± 4.7 Ab	49.2 ± 0.8 BCb	62.7 ± 6.2 ABde
400	50 ± 16.7 Ac	41.1 ± 4 ABb	39.1 ± 3.6 ABd	22.3 ± 4.3 Bd	43.1 ± 4.4 ABd	33.4 ± 0.5 ABb	40.9 ± 1.8 ABe
500	0 ± 0 Cd	0 ± 0 Cc	36.6 ± 0.5 Bd	46.7 ± 10.2 Bc	57.5 ± 8.4 ABcd	0 ± 0 Cc	74.7 ± 2.3 Acd

**Table 3 biology-12-01343-t003:** Shoot salt tolerance index of *S. salsa* seedlings grown from seeds treated with different NaCl concentrations. Each treatment had 16 biological replicates. Different lower-case letters indicate significant differences (*p* < 0.05, Tukey’s test) between treatments within the same provenance, while different upper-case letters indicate significant differences (*p* < 0.05, Tukey’s test) between provenances under the same NaCl concentration.

Concentration (mM)	S1	S2	S3	S4	S5	S6	S7
0	100 ± 0 Ab	100 ± 0 Aab	100 ± 0 Aa	100 ± 0 Abc	100 ± 0 Aa	100 ± 0 Aab	100 ± 0 Ac
50	126.7 ± 9 Bab	119.9 ± 1.4 Ba	107.6 ± 2.1 Ba	118.1 ± 3.2 Bab	107.2 ± 2.8 Ba	115.7 ± 12.7 Ba	165.5 ± 6.7 Aa
100	135.2 ± 12.4 Aa	113.4 ± 7.5 Aa	109.5 ± 2.8 Aa	129 ± 6.7 Aa	105.9 ± 3 Aa	102.6 ± 19.9 Aab	140.8 ± 7.9 Aab
200	107.2 ± 5.5 Bab	109.2 ± 6.2 Ba	106.8 ± 2.4 Ba	105.3 ± 2.2 Bbc	111.6 ± 1.5 Ba	73.4 ± 11.2 Cab	136.8 ± 7.8 Ab
300	105 ± 8.1 Bab	82 ± 4.9 BCbc	86.4 ± 3.5 BCb	86.7 ± 1.7 BCcd	68.6 ± 2.8 CDb	50 ± 16.7 Dbc	145.6 ± 4.8 Aab
400	48.2 ± 10.9 ABc	66.6 ± 4.5 ABc	78.8 ± 1.6 ABb	50.2 ± 7.3 Ae	64.3 ± 12.3 ABb	54.3 ± 0.5 ABbc	86.9 ± 4 Bc
500	0 ± 0 Cd	0 ± 0 Cd	49.5 ± 0.5 Bc	66.2 ± 6.9 Bde	92.7 ± 2.6 Aa	0 ± 0 Cc	99.6 ± 6.4 Ac

**Table 4 biology-12-01343-t004:** Percent germination recovery of *S. salsa* seeds in non-saline germination medium after NaCl pretreatment. Each treatment had four biological replicates. Different lower-case letters indicate significant differences (*p* < 0.05, Tukey’s test) between treatments within the same provenance, while different upper-case letters indicate significant differences (*p* < 0.05, Tukey’s test) between provenances under the same NaCl concentration.

Concentration (mM)	S1	S2	S3	S4	S5	S6	S7
0	0 ± 0 Abc	0 ± 0 Aab	0 ± 0 Aab	0 ± 0 Abc	0 ± 0 Aa	0 ± 0 Aa	0 ± 0 Aab
50	0 ± 0 Bbc	3.3 ± 3.3 Ab	0 ± 0 Bab	0 ± 0 Bbc	0 ± 0 Ba	0 ± 0 Ba	0 ± 0 Bab
100	7.8 ± 4.5 Ab	10.3 ± 2.4 Aab	7.5 ± 4.4 Aa	25 ± 10.4 Aab	0 ± 0 Ba	1.5 ± 1.5 Ba	8.5 ± 3.5 Aa
200	16.5 ± 3 Aab	12.5 ± 4.3 Aab	20 ± 4.3 Aa	9.5 ± 5.5 Ab	7.3 ± 7.3 Aa	0 ± 0 Aa	18.3 ± 1.3 Aab
300	35.8 ± 5.8 Bbc	8.5 ± 3.9 Cb	35.8 ± 4.3 Bab	61.3 ± 12.3 Ac	0 ± 0 Ca	5.5 ± 2.2 Ca	28.8 ± 5.9 Bab
400	40.5 ± 3.9 ACc	27.3 ± 0.9 BCa	50 ± 6.4 ABb	59.8 ± 5.6 Ac	10.5 ± 4.5 Ca	0 ± 0 Da	39.8 ± 7.1 ACb
500	38 ± 3.8 Bac	14 ± 6.8 Bab	52.5 ± 4 Ab	45.8 ± 4.3 Aac	7 ± 2.7 Ba	2 ± 1.2 Ba	35.3 ± 1.7 Ab

**Table 5 biology-12-01343-t005:** Effects of NaCl, provenance, and their interactions on *S. salsa* seed germination and seedling growth.

		Provenance	NaCl	Provenance × NaCl
Germination	d.f	6	6	36
	χ^2^	1496.38	311.11	158.97
	*p*	<0.001	<0.001	<0.001
Total germination	d.f.	6	6	36
	χ^2^	280.52	1020.26	167.26
	*p*	<0.001	<0.001	<0.001
Root length	d.f.	6	6	29
	F	33.38	69.72	2.25
	*p*	<0.001	<0.001	<0.001
Shoot length	d.f.	6	6	29
	F	58.39	36.25	3.25
	*p*	<0.001	<0.001	<0.001
Recovery germination percentage	d.f.	6	6	36
	χ^2^	473.92	180.19	117.76
	*p*	<0.001	<0.001	0.004

**Table 6 biology-12-01343-t006:** Percent relative germination of *S. salsa* seeds after treatment with different NaCl concentrations. Each treatment had four biological replicates. Different lower-case letters indicate significant differences (*p* < 0.05, Tukey’s test) between treatments within the same provenance, while different upper-case letters indicate significant differences (*p* < 0.05, Tukey’s test) between provenances under the same NaCl concentration.

Concentration (mM)	S1	S2	S3	S4	S5	S6	S7
0	100 ± 0 Aab	100 ± 0 Aab	100 ± 0 Aa	100 ± 0 Aa	100 ± 0 Aa	100 ± 0 Aa	100 ± 0 Aa
50	107.3 ± 4.3 ABa	128.1 ± 19.2 Aa	97 ± 4 ABab	83.6 ± 1.8 Bbc	94.1 ± 2.1 ABa	86.1 ± 7 Ba	95.4 ± 4.3 ABa
100	105.5 ± 4.5 ABab	116.7 ± 11.3 Aab	86.5 ± 1.6 Bb	90.1 ± 2 Bab	101.1 ± 1.1 ABa	86.1 ± 8.3 Ba	84.6 ± 1 Bab
200	91.3 ± 3.7 Bb	118.8 ± 7.1 Aa	86.3 ± 2.2 BCb	73.4 ± 3.3 Cc	98 ± 3 Ba	36.7 ± 1.9 Db	71.7 ± 2.5 Cbc
300	75.5 ± 2.7 c	91.7 ± 13.2 Aab	70.4 ± 3.1 Ac	76.1 ± 6.8 Abc	67.4 ± 3.6 Ab	33.9 ± 4.2 Bb	75.7 ± 4.7 Abc
400	64.8 ± 2.6 Bcd	149 ± 18.7 Aa	71.8 ± 4.8 Bc	70.1 ± 3.3 Bc	70.2 ± 3 Bb	4.2 ± 2.5 Cc	72.7 ± 5.5 Bbc
500	57.3 ± 3.5 Ad	50 ± 21.5 Ab	67.5 ± 1.5 Ac	52.6 ± 3.3 Ad	38.6 ± 5.2 ABc	4.2 ± 2.5 Bc	63.1 ± 2.8 Ac

**Table 7 biology-12-01343-t007:** Relative percent germination of S. salsa after treatment with different concentrations of Na2CO3. Each treatment had four biological replicates. Different lower-case letters indicate significant differences (*p* < 0.05, Tukey’s test) between treatments within the same provenance, while different upper-case letters indicate significant differences (*p* < 0.05, Tukey’s test) between provenances under the same Na2CO3 concentration.

Concentration (mM)	S1	S2	S3	S4	S5	S6	S7
0	100 ± 0 Aa	100 ± 0 Aa	100 ± 0 Aa	100 ± 0 Aa	100 ± 0 Aa	100 ± 0 Aa	100 ± 0 Aa
5	78.7 ± 5.8 ABbc	104.2 ± 4.2 Aa	45 ± 1.8 CDb	57.3 ± 9.9 BDb	32.7 ± 3.4 Db	62.5 ± 7 BCb	61.3 ± 6.4 BCb
12.5	93.7 ± 2.6 Aab	82.3 ± 7.7 Aab	28 ± 5.3 Cb	37.4 ± 8.5 BCbc	38 ± 1.9 BCb	48.9 ± 3.7 BCbc	52 ± 2 Bb
25	86.6 ± 2.6 Aac	57.3 ± 4.3 Bb	46.1 ± 8.2 BCb	45.7 ± 3.6 BCbc	22.5 ± 5.6 Cb	49.7 ± 3.7 BCbc	45.4 ± 10.7 BCb
50	77.1 ± 4.4 Ac	65.6 ± 6.9 ABb	32.3 ± 1.5 CDb	25.9 ± 5.7 Dc	26.6 ± 8.8 Db	56.7 ± 7.9 ACb	38.8 ± 7 BDb
100	74.1 ± 1.4 Ac	71.9 ± 8.6 Ab	47.6 ± 6.9 ABb	26.7 ± 6.5 Bc	29.3 ± 2 Bb	26.4 ± 10.5 Bc	42.9 ± 9.7 ABb

**Table 8 biology-12-01343-t008:** Root stress tolerance index in *S. salsa* seedlings grown from seeds treated with different Na_2_CO_3_ concentrations. Each treatment had 16 biological replicates. Different lower-case letters indicate significant differences (*p* < 0.05, Tukey’s test) between treatments within the provenance, while different upper-case letters indicate significant differences (*p* < 0.05, Tukey’s test) between provenances within Na_2_CO_3_ concentration.

Concentration (mM)	S1	S2	S3	S4	S5	S6	S7
0	100 ± 0 Ad	100 ± 0 Ad	100 ± 0 Ac	100 ± 0 Ab	100 ± 0 Ad	100 ± 0 Ab	100 ± 0 Ab
5	346 ± 15 Aa	312.7 ± 13.4 ABa	193.2 ± 9.2 Cab	238.4 ± 7.3 BCa	194.4 ± 4 Ca	260.4 ± 41 BCa	220.6 ± 10.3 Ca
12.5	291 ± 16.4 Ab	278.4 ± 8.4 ABa	208.3 ± 8.3 BDa	260.3 ± 21.7 ACa	197.1 ± 3.3 CDa	136.4 ± 23.3 Db	223.7 ± 19.3 aC
25	263.1 ± 15.4 Ab	233.1 ± 8.6 ABb	173.1 ± 8.4 CDb	230.9 ± 9.8 ACa	161.6 ± 3.5 Db	133 ± 23.5 Db	185.5 ± 11.6 BDa
50	160 ± 3.3 Ac	151.7 ± 5.6 ABc	115.9 ± 2.3 BCc	129.4 ± 8 ACb	115.5 ± 3.3 BDc	77.7 ± 14.8 Db	99.4 ± 11.9 CDb
100	65.3 ± 2.3 ABd	73.3 ± 1.8 Ad	45.1 ± 2.4 Cd	48.9 ± 2.6 BCc	64 ± 1.3 ACe	54.9 ± 9.2 ACb	47.1 ± 5.1 BCc

**Table 9 biology-12-01343-t009:** Shoot stress tolerance index of *S. salsa* seedlings grown from seeds treated with different Na2CO3 concentrations. Each treatment had 16 biological replicates. Different lower-case letters indicate significant differences (*p* < 0.05, Tukey’s test) between treatments within the same provenance, while different upper-case letters indicate significant differences (*p* < 0.05, Tukey’s test) between provenances under the same Na_2_CO_3_ concentration.

Concentration (mM)	S1	S2	S3	S4	S5	S6	S7
0	100 ± 0 Ab	100 ± 0 Ac	100 ± 0 Ac	100 ± 0 Ab	100 ± 0 Ab	100 ± 0 Aa	100 ± 0 Ab
5	104.9 ± 7.4 Cab	147.5 ± 4.1 Ab	114.3 ± 2.7 ACab	106.3 ± 2.6 BCb	107.7 ± 4 BCb	138.6 ± 16 ABa	141.2 ± 2.8 Aa
12.5	115.5 ± 8.2 Bab	173.7 ± 2.9 Aa	120.6 ± 3 Ba	120.4 ± 3 Ba	115.4 ± 6.6 Bb	136.6 ± 19 ABa	146.8 ± 10.3 ABa
25	114.9 ± 5.7 BCab	175.8 ± 3.7 Aa	105.3 ± 3.1 Cbc	121.2 ± 3.8 BCa	116.3 ± 3 BCb	109.2 ± 10.5 Ca	141.5 ± 7.5 Ba
50	132.7 ± 6.8 BCa	179.4 ± 0.9 Aa	116 ± 0.7 BCa	125.4 ± 2.9 BCa	136.9 ± 3.4 BCa	106.2 ± 14.8 Ca	146.1 ± 10.9 ABa
100	130.8 ± 9.3 Bab	171.2 ± 4.6 Aa	116.9 ± 2.7 BCa	125.7 ± 1.5 Ba	137.7 ± 3.8 Ba	89.4 ± 12.8 Ca	143.3 ± 7.3 ABa

**Table 10 biology-12-01343-t010:** Effects of Na_2_CO_3_, provenance, and their interactions on *S. salsa* seed germination and seedling growth.

		Provenance	Na_2_CO_3_	Provenance × Na_2_CO_3_
Germination	d.f	6	5	30
	χ^2^	736.86	319.25	176.57
	*p*	<0.001	<0.001	<0.001
Total germination	d.f.	6	5	30
	χ^2^	628.11	3416.47	187.63
	*p*	<0.001	<0.001	<0.001
Root length	d.f.	6	5	30
	F	192.18	447.42	11.31
	*p*	<0.001	<0.001	<0.001
Shoot length	d.f.	6	5	30
	F	181.04	32.36	3.66
	*p*	<0.001	<0.001	<0.001
Recovery germination percentage	d.f.	6	5	30
	χ^2^	193.25	142.3	93.36
	*p*	<0.001	<0.001	0.016

**Table 11 biology-12-01343-t011:** Percent germination recovery in *S. salsa* seeds in non-saline germination medium after Na_2_CO_3_ pretreatment. Each treatment had four biological replicates. Different lower-case letters indicate significant differences (*p* < 0.05, Tukey’s test) between treatments within the same provenance, while different upper-case letters indicate significant differences (*p* < 0.05, Tukey’s test) between provenances under the same Na_2_CO_3_ concentration.

Concentration (mM)	S1	S2	S3	S4	S5	S6	S7
0	0 ± 0 Aa	0 ± 0 Aa	0 ± 0 Aab	0 ± 0 Ab	0 ± 0 Aa	0 ± 0 Aa	0 ± 0 Ab
5	3.5 ± 2 Ba	0 ± 0 Ba	4.8 ± 3.1 Bb	24.3 ± 6 Aa	6.8 ± 1.4 Ba	0 ± 0 Ba	25.3 ± 3.9 Aa
12.5	3.3 ± 3.3 Aa	0 ± 0 Aa	1.3 ± 1.3 Ab	2 ± 2 Ab	5 ± 3.3 Aa	0 ± 0 Aa	17 ± 2.5 Aa
25	6.8 ± 2.3 Aa	0 ± 0 Aa	8.5 ± 4.4 Ab	7.3 ± 5.2 Ab	1.3 ± 1.3 Aa	0 ± 0 Aa	18 ± 8.2 Aa
50	2 ± 2 Aa	0 ± 0 Aa	6.3 ± 0.9 Ab	5.5 ± 2.5 Ab	0 ± 0 Aa	0 ± 0 Aa	13.8 ± 3 Aa
100	5.8 ± 1.9 Ba	1.3 ± 1.3 Ba	30.3 ± 6.8 Aa	7.5 ± 3 Bb	6.3 ± 2.4 Ba	0 ± 0 ABa	19.5 ± 4.4 ABa

**Table 12 biology-12-01343-t012:** Effects of NaHCO_3_, provenance, and their interactions on *S. salsa* seed germination and seedling growth.

		Provenance	NaHCO_3_	Provenance × NaHCO_3_
Germination	d.f	6	4	24
	χ^2^	676.09	273.27	112.49
	*p*	<0.001	<0.001	<0.001
Total germination	d.f.	6	4	24
	χ^2^	534.47	253.52	111.55
	*p*	<0.001	<0.001	<0.001
Root length	d.f.	6	4	24
	F	204.4	262.89	7.76
	*p*	<0.001	<0.001	<0.001
Shoot length	d.f.	6	4	24
	F	129.55	46.22	5.1
	*p*	<0.001	<0.001	<0.001
Recovery germination percentage	d.f.	6	4	24
	χ^2^	141.99	90.54	61.19
	*p*	<0.001	<0.001	0.207

**Table 13 biology-12-01343-t013:** Root stress tolerance index in *S. salsa* seedlings grown from seeds treated with different NaHCO_3_ concentrations. Each treatment had 16 biological replicates. Different lower-case letters indicate significant differences (*p* < 0.05, Tukey’s test) across treatments within the same provenance, while different upper-case letters indicate significant differences (*p* < 0.05, Tukey’s test) between provenances under the same NaHCO_3_ concentration.

Concentration (mM)	S1	S2	S3	S4	S5	S6	S7
0	100 ± 0 Ac	100 ± 0 Ad	100 ± 0 Ac	100 ± 0 Ac	100 ± 0 Ac	100 ± 0 Aa	100 ± 0 Ac
25	296.6 ± 19.8 Aa	285 ± 2.5 Aa	226.9 ± 8.1 ABa	244.2 ± 20.4 ABa	195.3 ± 3.5 BCa	124.6 ± 24 Ca	240.3 ± 16 ABa
50	287.5 ± 4.4 Aa	237.7 ± 3.6 ABb	202.3 ± 10.9 BCa	218.3 ± 10.4 BCa	181.1 ± 6.1 Ca	115.2 ± 19.8 Da	197.8 ± 11.9 BCab
100	188.1 ± 7.2 Ab	161 ± 5.7 ABc	141.5 ± 6.5 Bb	154.8 ± 9.2 ABb	135.7 ± 2.1 Bb	71.3 ± 9.3 Ca	146.6 ± 12.6 Bbc
150	188.1 ± 7.2 Ab	161 ± 5.7 ABc	141.5 ± 6.5 Bb	154.8 ± 9.2 ABb	135.7 ± 2.1 Bb	71.3 ± 9.3 Ca	146.6 ± 12.6 Bbc

**Table 14 biology-12-01343-t014:** Shoot stress tolerance index in *S. salsa* seedlings grown from seeds treated with different NaHCO_3_ concentrations. Each treatment had 16 biological replicates. Different lower-case letters indicate significant differences (*p* < 0.05, Tukey’s test) among different NaHCO_3_ concentrations within the same provenance, while different upper-case letters indicate significant differences (*p* < 0.05, Tukey’s test) between provenances under the same NaHCO_3_ concentration.

Concentration (mM)	S1	S2	S3	S4	S5	S6	S7
0	100 ± 0 Ab	100 ± 0 Ab	100 ± 0 Ab	100 ± 0 Ab	100 ± 0 Ab	100 ± 0 Aa	100 ± 0 Ab
25	117.6 ± 8.8 BCab	179.3 ± 5.4 Aa	121.1 ± 3.3 BCa	104.1 ± 1.3 Cab	106.1 ± 8.2 Cb	116.8 ± 16.9 BCa	153.5 ± 6.4 ABa
50	127.6 ± 5.9 BCa	177.3 ± 4.4 Aa	122.4 ± 3.1 BCa	102.7 ± 3.6 Cab	120.1 ± 5.8 Cb	139.1 ± 22.3 ACa	166.3 ± 6.9 ABa
100	134.6 ± 6.1 BCa	182.6 ± 7.4 Aa	123.7 ± 5.7 BCa	116.6 ± 4.5 Ca	153.8 ± 4.9 ABa	126.7 ± 15.2 BCa	148.4 ± 7.1 ACa
150	134.6 ± 6.1 BCa	182.6 ± 7.4 Aa	123.7 ± 5.7 BCa	116.6 ± 4.5 Aa	153.8 ± 4.9 ABa	126.7 ± 15.2 BCa	148.4 ± 7.1 ACa

**Table 15 biology-12-01343-t015:** Percent germination recovery of *S. salsa* in non-saline germination medium after NaHCO_3_ pretreatment. There were four biological replicates in each treatment. Different lower-case letters indicate significant differences (*p* < 0.05, Tukey’s test) in germination recovery among NaHCO_3_ treatments within the same provenance, while different upper-case letters indicate significant differences (*p* < 0.05, Tukey’s test) between provenances under the same NaHCO_3_ concentration.

Concentration (mM)	S1	S2	S3	S4	S5	S6	S7
0	0 ± 0 Aa	0 ± 0 Aa	0 ± 0 Aa	0 ± 0 Aa	0 ± 0 Aa	0 ± 0 Aa	0 ± 0 Aab
25	5 ± 2.9 Aa	0 ± 0 Aa	7 ± 2.6 Aa	2.8 ± 2.8 Aa	5.3 ± 2.1 Aa	0 ± 0 Aa	14.3 ± 4 Aa
50	7.5 ± 4.8 Ba	0 ± 0 ABa	5.3 ± 2.1 Ba	8.3 ± 2.6 Ba	13.5 ± 1.8 Ba	0 ± 0 ABa	40.5 ± 4.3 Ab
100	2.5 ± 2.5 Ba	0 ± 0 Ba	7.8 ± 3.7 Ba	5.3 ± 2.1 Ba	2.3 ± 1.3 Ba	0 ± 0 Ba	39.3 ± 3.6 Ab
150	7.8 ± 2.9 Ba	2.3 ± 1.3 Ba	12.5 ± 1.7 Ba	14.5 ± 3.7 ABa	1.3 ± 1.3 Ba	0 ± 0 Ba	34.3 ± 6.9 Ab

**Table 16 biology-12-01343-t016:** Relative percent germination of *S. salsa* seeds after treatment with different NaHCO_3_ concentrations. Each treatment had four biological replicates. Different lower-case letters indicate significant differences (*p* < 0.05, Tukey’s test) across treatments within the same provenance, while different upper-case letters indicate significant differences (*p* < 0.05, Tukey’s test) between provenances under the same NaHCO_3_ concentration.

Concentration (mM)	S1	S2	S3	S4	S5	S6	S7
0	100 ± 0 Aa	100 ± 0 Aab	100 ± 0 Aa	100 ± 0 Aa	100 ± 0 Aa	100 ± 0 Aa	100 ± 0 Aa
25	84.6 ± 4 Ac	103.1 ± 7.9 Aa	41.2 ± 0.8 Bb	37.3 ± 0.9 Bb	30 ± 4.5 Bb	26.9 ± 5.3 Bd	43.1 ± 6.5 Bc
50	96.8 ± 1.9 Aab	59.4 ± 19.9 ACb	39.1 ± 4.9 Cb	38.3 ± 1.4 Cb	41.9 ± 0.6 Cb	53.3 ± 7.2 BCbc	81.3 ± 4.5 ABab
100	93.6 ± 3 Aac	70.8 ± 2.4 Bab	21.8 ± 4.9 Cc	26.1 ± 4 Cc	27.1 ± 6.5 Cb	35 ± 4.8 Ccd	74.5 ± 3.7 ABb
150	86.8 ± 2.3 Abc	74 ± 4.9 ABab	28.5 ± 1.7 Cbc	34.9 ± 2.8 Cbc	23.4 ± 6.5 Cb	62.5 ± 7 ABb	59.9 ± 8.2 Bbc

## Data Availability

The data presented in this study are available in the article.

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
