# Peer review of "Seed Germination and Seedling Growth in Suaeda salsa (Linn.) Pall. (Amaranthaceae) Demonstrate Varying Salinity Tolerance among Different Provenances"

_biology, 2023, doi:10.3390/biology12101343_

Round 1
Reviewer 1 Report
The article has scientific interest, well structured, and congruent among its different sections; though, some aspects need to be improved:
i) Scientific names, it is important to write the full name, and to add the Authority and the family, the first time it appears in the document; subsequently, it can be used the abbreviation of the genus (S. salsa); however, when the scientific name starts a paragraph, it has to be full written (Suaeda salsa). All scientific names are written in Italics.
ii) All tables and figures need a Title, and a clear, precise and explanatory Legend. Tables and figures ought to be completely understandable for the reader by themselves, without reading the text; and all the abbreviations are explained.
iii) Material and Methods, there are two huge gaps in this section:
a) Need to clearly explain the number of seeds and seedlings used in each treatment, including the control.
b) Considering the objectives of this research (soil salinity tolerance), it is difficult to understand why none soil analysis was done. In fact, the physico-chemical analysis of the soil would have given key information for comparing among the 7 sites (seeds´ provenances) studied, as well as the seed germination and seedling formation differences.
The crucial point in this study is the soil (saline soil); mainly to understand how the different plant ecotypes (seeds and seedlings) are adapted to their specific habitat. In fact, this ecological point of few is lacking in the Discussion.
iv) Discussion, 11 references are directly (8 references,) and indirectly (3 references) related to the objectives of the research.
References 12, 17, 19, 20, 22, 23, 27 and 28, are related to seed germination and seedling growth of Suaeda salsa in saline soils; references 21, 24 and 26, are related to S. salsa cytology, hydrological characteristics and recruitment, and climate change in saline soils.
Hence, the question here is to know what is the novelty of this study in relation to the other ones done before?
Please, see file attached.

Reviewer 2 Report
Please see the attachment.

There are some grammatical errors that need to be corrected.
Reviewer 3 Report
Dear authors,
I appreciate the opportunity to review your manuscript, but in the current form, I do not recommend publishing since there are several issues - major issues are listed below and in the PDF file.
1) First of all, I would like to say that there are numerous issues with grammar and style in the text in many cases I had to guess what the authors wanted to say. I began to mark in the text fragments that have to be checked and corrected, but once I realized that they were a lot, I gave up. The authors should consult a native speaker or can use InstaText, Grammarly, or any other linguistic program useful for checking English. These programs are not ideal, but provide some suggestions for text improving
2) There is a Lack of punctuation, verbal time is not the same throughout and some species names are not in italic.
3) Methodology is not detailed and does not describe the methods enough.
4) Results are extensive and somehow confusing and exhausting. You should rethink the way you display the results.
5) The results are barely discussed, and the main point that lacks for me is the soil environmental data from the different provenance.

English is very difficult to understand/incomprehensible.
Round 2
Reviewer 1 Report
Authors have done all the corrections suggested; however, few a pair of small corrections are asked to correct.
As well, authors need to focus on the References section; ought to standardize the use of capital and small letters.
Please, see file attached.
